# Treatment burden in individuals living with and beyond cancer: A systematic review of qualitative literature

Rosalind Adam[1]*, Revathi Nair[2], Lisa F. Duncan[1], Esyn Yeoh[2], Joanne Chan[2], Vaselisa Vilenskaya[2], Katie I. Gallacher[3]

1 Institute of Applied Health Sciences, University of Aberdeen, Aberdeen, United Kingdom, 2 School of Medicine and Dentistry, University of Aberdeen, Aberdeen, United Kingdom, 3 Institute of Health & Wellbeing, General Practice & Primary Care, University of Glasgow, Glasgow, United Kingdom

* rosalindadam@abdn.ac.uk

**Data Availability Statement:** Review of published literature - all original articles are available online.

**Funding:** This research was funded by the Chief Scientist Office (https://www.cso.scot.nhs.uk/)

## Abstract

### Background

Individuals with cancer are being given increasing responsibility for the self-management of their health and illness. In other chronic diseases, individuals who experience treatment burden are at risk of poorer health outcomes. Less is known about treatment burden and its impact on individuals with cancer. This systematic review investigated perceptions of treatment burden in individuals living with and beyond cancer.

### Methods and findings

Medline, CINAHL and EMBASE databases were searched for qualitative studies that explored treatment burden in individuals with a diagnosis of breast, prostate, colorectal, or lung cancer at any stage of their diagnostic/treatment trajectory. Descriptive and thematic analyses were conducted. Study quality was assessed using a modified CASP checklist. The review protocol was registered on PROSPERO (CRD42021145601). Forty-eight studies were included. Health management after cancer involved cognitive, practical, and relational work for patients. Individuals were motivated to perform health management work to improve life-expectancy, manage symptoms, and regain a sense of normality. Performing health care work could be empowering and gave individuals a sense of control. Treatment burden occurred when there was a mismatch between the resources needed for health management and their availability. Individuals with chronic and severe symptoms, financial challenges, language barriers, and limited social support are particularly at risk of treatment burden. For those with advanced cancer, consumption of time and energy by health care work is a significant burden.

### Conclusion

Treatment burden could be an important mediator of inequities in cancer outcomes. Many of the factors leading to treatment burden in individuals with cancer are potentially modifiable.

Scottish Clinical Academic Fellowship (Grant CSO-SCAF/18/02). This grant was awarded to RA. The funder had no role in the study design, data collection and analysis, decision to publish, or preparation of the manuscript.

**Competing interests:** The authors have declared that no competing interests exist.

Clinicians should consider carefully what they are asking or expecting patients to do, and the resources required, including how much patient time will be consumed.

## Introduction

There is considerable and growing interest in the negative impact of treatment burden on health outcomes in chronic diseases [1–3]. Treatment burden is the workload of health care for people with chronic illness, and the impact that this has on functioning and well-being [4]. Individuals who become over-burdened by the workload of health care may disengage from self-management practices, leading to poorer patient outcomes [4–7]. Treatment burden is salient in the oncology setting as the responsibility for illness management and health recovery is shifting away from healthcare systems toward self-management by patients and their families [8–11].

Treatment burden research has particularly focused on "chronic diseases" such as cardiovascular diseases [5, 6, 12, 13], diabetes [14], and in multimorbidity [15–17]. Cancer is increasingly being recognised as a chronic disease, and treatment burden is likely to be of considerable importance in cancer [18]. Questionnaire studies have shown that individuals with multimorbidity, low levels of social support, and low health literacy may experience high levels of treatment burden after cancer [19–21]. It is plausible that treatment burden could be an important mediator of inequities in cancer outcomes, for example in individuals with reduced capacity for self-management due to socioeconomic deprivation [22] and in rural patients who face long travelling times [23].

The aim of this review is to investigate patient perceptions of treatment burden after a cancer diagnosis and to explore the impact of treatment burden on individuals living with and beyond cancer. The review focuses on breast, prostate, colorectal, and lung cancers because they are the most common cancers globally [24], and encompass a range of symptoms, prognoses, treatment modalities, and late effects.

## Methods

A systematic review of qualitative literature was conducted to identify, characterise, and explore lived experiences and perceptions of treatment burden in individuals living with and beyond breast, colorectal, prostate and lung cancer. The protocol for the review was registered on PROSPERO (registration number: CRD42021145601). An ENTREQ reporting checklist [25] for this systematic review is available in S1 Table.

### Identification of studies

Treatment burden is a relatively recent concept, and it was considered unlikely at the outset that many studies would explicitly use the terms "treatment burden" or "burden of treatment" in the title, abstract, or index. To overcome this challenge, treatment burden theoretical models [4, 7, 13, 26], conceptual reviews [3, 6, 13, 18, 27], and measurement scales [16, 17, 28] were examined to identify examples of "work" or self-management behaviours that could be incorporated into database searches. A database search strategy (S1 Appendix) was devised in consultation with a senior medical librarian that encompassed terms relating to cancer, self-management behaviours, and qualitative research. Several iterations of the search strategy were tested, and search terms were refined to return studies relevant to the review question. Medline, CINAHL and EMBASE databases were searched from the year 2000 onwards.

**Table 1. Inclusion and exclusion criteria.**

| Inclusion | Exclusion |
|---|---|
| English language | Literature reviews or syntheses, unpublished work, letters |
| Publication date 2000 to current | Studies evaluating the effect of a research intervention or clinical trial |
| Individuals with a confirmed diagnosis of colorectal, lung, prostate, or breast cancer, at any point in disease and treatment trajectory. Studies including patients with other cancers were included if they also included participants with one of the four cancer types listed. | Accounts of cancer screening or diagnostic activities in individuals without a confirmed cancer diagnosis |
| Includes behaviours/work taken to self-manage health after cancer **AND** the impact that this has had on the individual in terms of function and/or wellbeing (treatment burden) | Only describes burden of illness such as experience of cancer, cancer symptoms, side effects, late effects of treatment, psychological adjustment or coping response without describing the work of self-management activities. |
| Qualitative research examining **patient** perceptions of treatment burden and the impact of treatment burden on this same individual | Examines behaviours/work of self-management without exploring the impact this has on the individual |
| Includes raw data in the form of participant quotations | Research that only includes caregivers or healthcare professionals, or that seeks to examine the effect of cancer on others without a cancer diagnosis (e.g., work of caring for an individual with cancer, effect of cancer on spouse/family) |
| Mixed methods research that has a discrete qualitative component and meets all the other inclusion criteria | |

Cancer treatments have evolved over time, and it was important that patient experiences reflected modern cancer treatment pathways.

Inclusion and exclusion criteria are summarised in Table 1.

Titles and abstracts were imported into Proquest Refworks (https://refworks.proquest.com/) and duplicates removed. Titles and abstracts were reviewed independently by two authors (RA, JC, AK, EY, and VV). Full texts were retrieved for all studies which were judged by at least one author to be potentially eligible for inclusion. Full texts were reviewed independently by at least two authors and any disagreements were resolved by discussion with a third author.

## Data extraction and synthesis

A data extraction form was designed in Microsoft Word to capture descriptive data about the paper, the key themes, messages, and main findings described by the original study authors. Verbatim participant quotations about treatment burden were also extracted. Data extraction was performed independently by at least two authors, and three authors (RA, LD, and AK) met to agree the final data extraction form for each study. A unified document containing the data extracted from all studies was circulated to the whole review team.

Study characteristics were summarised using descriptive statistics. Thematic synthesis [29] was used to synthesise qualitative study data. All authors met to discuss key findings and concepts that were present across multiple studies. Codes (short descriptive terms) were agreed, which represented meaningful concepts present across the studies. The lead author coded the extracted study data in NVivo version 12, noting similarities, differences, and relationships between the codes. Analytical themes were generated, discussed, and refined within the whole review team. The studies were also examined for any similarities or differences in treatment burden between different cancer types. Analytical themes (similar to third order interpretations in meta-ethnography [30]) remained true to the findings of the original studies but went

beyond the original research to generate additional insights into concepts of treatment burden after cancer.

### Quality assessment

Study quality assessment was undertaken independently by two authors (JC and EY) using a modified CASP checklist [31]. Areas of uncertainty or disagreement were resolved by discussion with a third author (LD). Studies were not excluded based on quality assessment.

### Ethical considerations

No participant quotations were used in this review from any study without an explicit statement of ethical approval, and all quotations used in this review were published in a peer reviewed journal and already in the public domain.

## Results

A PRISMA chart is shown in Fig 1. Database searches were carried out in June 2021 and returned 14,730 titles. Two additional records were identified through other sources. After 7375 duplicate titles were removed there were 7357 titles and abstracts. Of those, 119 full texts were judged to be potentially relevant to the review. In total, 45 original research articles met the inclusion criteria. An updated Medline search was carried out in May 2022. Three additional articles met the inclusion criteria, resulting in 48 original research articles in the final review.

### Characteristics of included studies

Descriptions of the 48 original qualitative research articles, their study population and scope are given in S2 Table. Studies included individuals with breast cancer (n = 19, 39.6%) [32–50], colorectal cancer (n = 13, 27.1%) [51–63], lung cancer (n = 5, 10.4%) [64–68], and prostate cancer (n = 2, 4.2%) [69, 70]. Nine studies (18.8%) included individuals with a mix of cancer types [71–79]. The studies were conducted in the United States (n = 13, 27.1%) [33–35, 38, 45, 46, 48, 49, 63, 67, 72, 73, 79], UK or Ireland (n = 11, 22.9%) [36, 41, 51–53, 56, 71, 75–78], Canada (n = 7, 14.6%) [39, 50, 54, 61, 65, 69, 70], Taiwan (n = 4, 8.3%) [42, 57, 58, 66], Sweden (n = 4, 8.3%) [32, 55, 60, 74], China (n = 2, 4.2%) [37, 68], and Germany (n = 2, 4.2%) [40, 59]. The remaining five studies were conducted in Australia, Turkey, Thailand, Denmark, and Indonesia (each n = 1) [43, 44, 47, 62, 64].

The studies included 1250 participants (median 16, range 3 to 178 participants), and collected qualitative data through interviews (n = 33, 68.8%) [36–42, 44, 46–50, 52, 53, 55–58, 60–62, 64–68, 72–75, 77, 78], focus groups (n = 6, 12.5%) [32, 33, 45, 51, 54, 59], free text written responses (n = 1, 2.1%) [63], or a combination of approaches such as interviewing and focus groups, integration of diary, drawing methods, and free text responses (n = 8, 16.7%) [34, 35, 43, 69–71, 76, 79].

### Analytical themes

The analytical themes and their relationships are summarised in Fig 2.

### Theme one: The work of cancer and cancer management

Cancer management was compared to or described as "work" by participants. The wide-ranging work of cancer management is summarised in Table 2. The cancer experience was time-

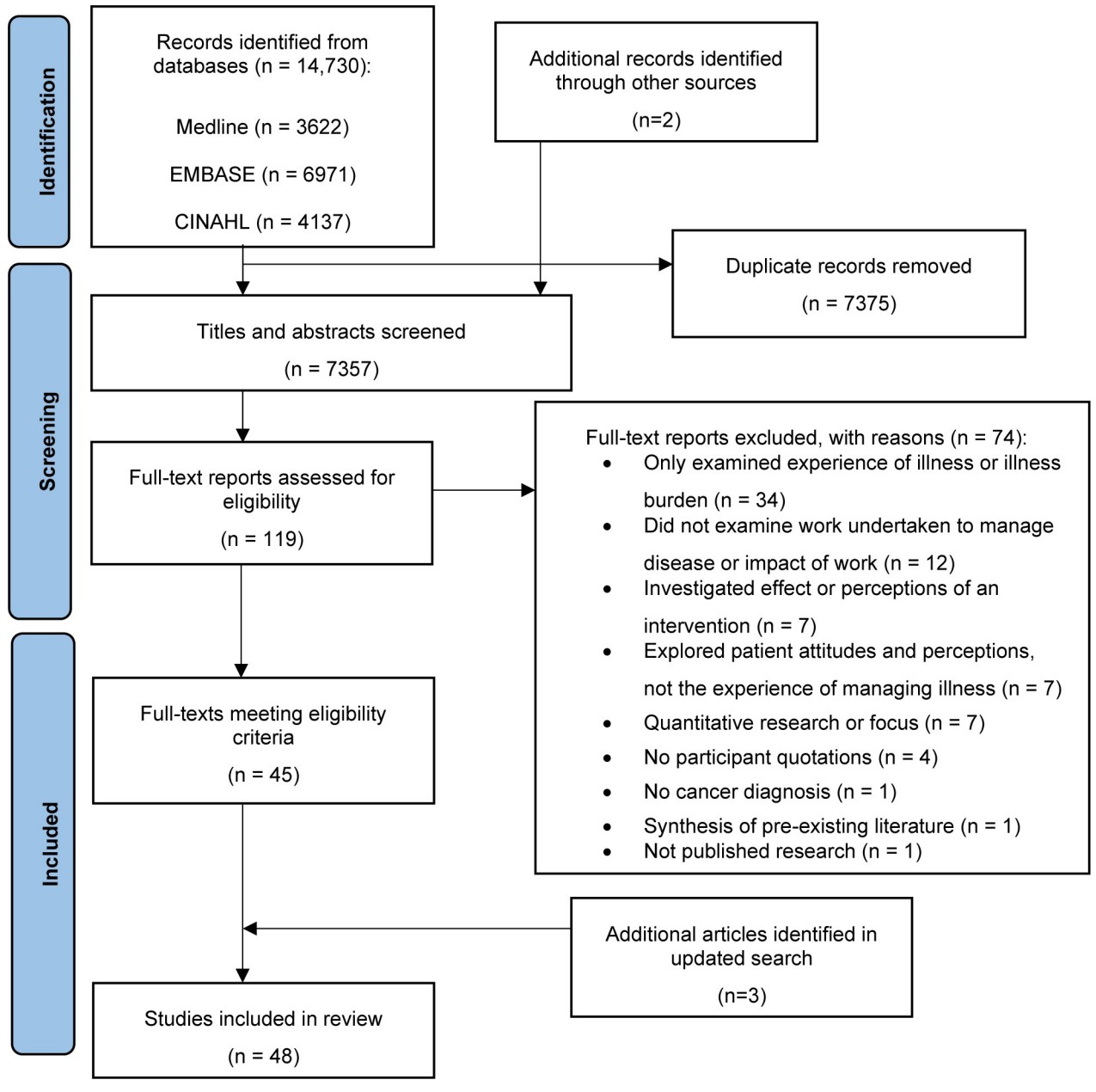

**Fig 1. PRISMA flow diagram of study inclusion process.**

consuming and included cognitive work, practical work, and relational work. Participant quotations exemplifying the work of cancer management are presented in Table 3.

The notion of cognitive work, or time spent thinking about cancer management, was present across most studies. Cancer was always present in the background, and effort was expended trying to engage in other meaningful activities to distract from cancer or to maintain a "positive mental attitude".

Cognitive work also involved sense-making activities, such as reading about cancer or its treatment and assimilating personal medical records. Participants in several studies described additional time spent planning their activities to account for problematic symptoms or

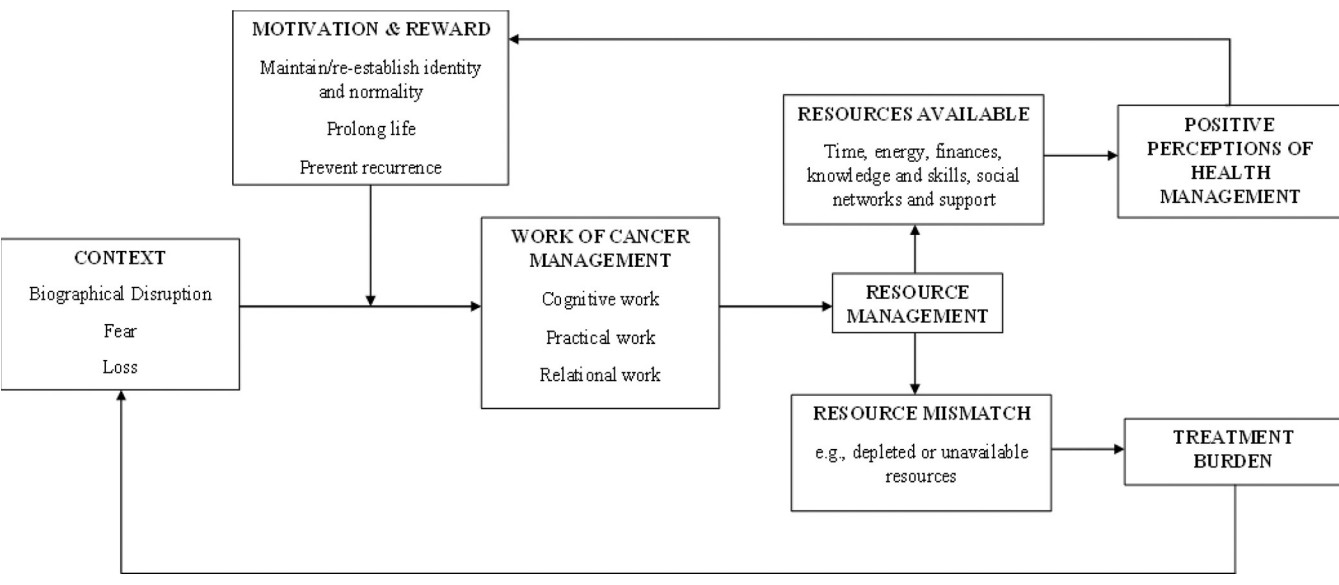

**Fig 2. Analytical themes and relationships between them.**

potential complications, for example, thinking about the location of toilets for those with unpredictable bowel symptoms or incontinence, or carrying medicines with them. Financial challenges associated with or caused by cancer and its treatment also required significant planning and administrative effort.

Individuals with cancer engaged in the practical work of attending appointments, taking medicines regularly, and managing symptoms. Examples included using wraps or manual lymphatic drainage to manage lymphoedema or making up weekly pill boxes to improve medication adherence. Many tasks involved a combination of practical and cognitive work, for example, concealing a stoma might involve planning what to wear, and then the practical tasks of using tape to mask the stoma. Several studies described participants making efforts to avoid the topic of cancer being raised by others during social or daily activities. Individuals were motivated to conceal physical signs of their cancer treatment to maintain a sense of identity and normality, to maintain their sense of physical attractiveness, and to mitigate social stigma or judgements about having cancer.

Relational work for those with cancer involved enlisting the support of others (such as healthcare professionals, friends, and significant others) to assist with the work of health management. Many studies [34, 37, 46, 47, 49, 59, 62, 65, 69, 76, 78] noted that cancer and cancer treatment altered the dynamics of personal relationships by reducing sexual intimacy or changing roles within relationships. For example, friends and loved ones had to take on caregiving roles, and individuals with cancer struggled to fulfil the family responsibilities they had valued prior to their diagnosis (e.g., looking after children, helping with household chores). It took significant effort to maintain important relationships during cancer treatment and beyond.

## Theme two: Context, motivation and reward

A recurring finding across most studies was that cancer caused significant biographical disruption. It was not only a threat to life, but also threatened the individual's sense of identity and current way of life, altering their sense of "normality". Fear was a common response to cancer, including fear of death and fear of recurrence in cancer survivors. Cancer was associated with

**Table 2. The cognitive, practical and relational work of living with and beyond cancer.**

| Nature of health-related work | Self-management work or behaviour | Example(s) noted across literature | Perceived impact, consequences, or results of the work |
|---|---|---|---|
| Cognitive work | Gaining knowledge about cancer and its management | Searching for information from a range of sources, determining the relevance and reliability of the information | Making sense of cancer and its treatment<br>Increased sense of control<br>Self-advocacy and increased engagement in medical consultations<br>Time spent in pursuit of knowledge about disease reduces time available for other meaningful activities |
| | Management of distress, fear of recurrence or other psychological symptoms | Actively distracting self from cancer by engaging in other activities<br>Downward comparison with others "worse off"<br>Visualisation, relaxation, maintaining positive attitude | "Coping" with cancer, reducing "obsession" or recurring thoughts about cancer, reducing distress and fear about cancer and its implications |
| Practical and cognitive work | Medicines management | Ordering prescriptions, adhering to a medication regime, pre-planning e.g., carrying supplies of medicines to deal with complications, such as antibiotics for cellulitis in lymphoedema, filling pill boxes | Medicines management perceived to be important for disease control, extending life (e.g., epidermal growth factor inhibitors in lung cancer), preventing recurrence (e.g., adjuvant hormonal treatments in breast cancer), controlling symptoms and side effects of cancer treatment |
| | Management of symptoms and immediate or late effects of treatment | Pre-planning e.g., location of toilets for bowel/continence symptoms, medicines required for travel<br>Self-monitoring, e.g., monitoring relationship between food intake and bowel symptoms, monitoring arm circumference in lymphoedema to tailor self-management activities<br>Pacing activities. Making notes/reminders to manage cognitive effects of anti-cancer treatment | Reducing the intensity, severity, or impact of the symptoms on daily life. |
| | Managing the physical changes of cancer/cancer treatment | Carefully choosing clothing to conceal stoma or lymphoedema, using tape, bandages, or wrapping | Maintain sense of identity and normality<br>Maintain sense of physical attractiveness<br>Avoid the topic of cancer coming up during social or daily activities<br>Manage stigma of cancer |
| | Enacting lifestyle changes | Exercising, making modifications to the diet, quitting smoking or reducing alcohol consumption | Manage physical symptoms, for example, modifying diet to control bowel symptoms after colorectal cancer surgery<br>Improved quality of life and sense of well-being<br>Reduce risk of recurrence/improve disease outcomes |
| | Living life with/after cancer | Reprioritising to achieve goals or altered priorities after cancer diagnosis<br>Managing finances<br>Participating in or maintaining career or employment activities<br>Engaging in spiritual practices<br>Activities of daily living, such as housework, self-care | Maintaining normality<br>Managing the financial burden of cancer<br>Spiritual practices help with psychological adjustment and were perceived to be life enhancing and important in improving disease outcomes. |
| Cognitive and relational work | Interacting with the healthcare system | Navigating care systems, arranging appointments, planning for appointments, travelling to healthcare appointments<br>Self-advocacy–"pushing" or "actively" seeking second opinions, consultations and taking an active role in decision making | Sense of control<br>Disease management, improving disease outcomes, sense of reassurance.<br>Time spent interacting with and attending healthcare facilities reduces time available for other meaningful activities |
| Relational work | Maintaining and modifying personal and professional relationships | Seeking formal support (e.g., from support groups/ organisations) and informal support from others, consulting with professionals, participating in social interactions<br>Maintaining family relationships | Improved psychological well-being and quality of life<br>Delegation of self-management activities reduces work |

loss, including loss of physical functioning, loss of independence/increased reliance on others, loss of meaningful activities, financial loss, and loss of social relationships or changed relationship dynamics (exemplar quotations are in Table 3). Fear, the wish to prolong life or prevent

**Table 3. Exemplar quotations illustrating the analytical themes.**

| Analytical Theme | Finding | Exemplar Quotations, with participant demographic details (where available) |
|---|---|---|
| Theme one: The work of cancer and cancer management | Treatment is compared to work | "(When I) have these appointments I have to go to, I just pencilled it in... and (it has to) just be like, this is part of my job right now is, taking care of my health and because I treated it that way, it was so much easier to deal with that. It was just something that needed to be done." [from Sun [49], female, breast cancer survivor with lymphoedema (age missing)] |
| | Treatment is time consuming, particularly during acute treatment | "At the rehabilitation facility, I was told not to take on too many [working] hours too quickly, if possible, to make sure that I would have enough time for all my (laughs) appointments, that is, the exercise and doctor's appointments because that just remains really time-intensive for the first few years. [. . .] Afterward, you continue to have mammography and gynecologist appointments. I had to keep going to radiotherapy. Then there are other minor ailments that results from this chemotherapy, organs that might have cause some problems, teeth and eyes. (Laughs). Everything suffered a little under it. So, I had a lot of appointments in the first few years after acute therapy. And so, I simply had to make sure that I kept two days a week open. One day for rest and housework. And one, on average, for doctor's appointments. Actually, I have stuck to that to this day." [from Hiltrop [40], female, breast cancer survivor (age missing)] |
| Theme two: context, motivation and reward | The work of health management takes place in the context of significant biographical disruption and illness burden | "We have not felt normal since we suffered from this illness (cancer). What can we do for survival? We have been mentally exhausted. We can't live without medicine. We need regular exercise. We can't go outing because we need to take herbal medicine twice a day. We can't also eat outside because foods are unsafe and unhealthy. We have no choice if we get this illness." [from Cheng [37], 49-year-old female, breast cancer survivor] |
| | Fear, the wish to prolong life or prevent recurrence, and the desire to maintain or re-establish personal identity/sense of normality are strong motivating factors | "[Self-care is] the immediate things like taking care of my body and making sure that I do everything I possibly can with hygiene, diet, all possible these things to keep me as fit as...it's got a wider aspect, it's about your mental state as well I think and trying to be as normal as possible and trying to just be you...it's a whole big thing, it's not just the physical...it's the mental side of it as well and just trying to keep going and be the person I always was." [from Kidd [56], 49-year-old female with colorectal cancer] |
| | | "poop in a plastic bag. . ..or lie in a pine box" [from McMullen [63], colorectal cancer survivor (sex and age missing)] |
| Theme three: resource management | Social support and social networks are important personal resources. Work can be delegated to others. | "My husband was so supportive of my treatment that he had to leave his job. He used to work at the airport, but because of taking care of me, he quit his job and is now working non- permanent jobs. . . as he'd need to accompany my visit to the hospital 2–3 times a week, for chemotherapy, routine check-ups, and picking up the medication." [from Prabandari [44], 46-year-old female with advanced breast cancer] |
| | | "Somebody's driving me [to clinic appointments], at the moment. . .so I go all the way round my friends, they're all doing one or two, so it's not one person doing it, because it's unfair, that." [from Walshe [76], advanced cancer (sex, age, and cancer type missing)] |
| | "Energy" is an important resource that requires careful management | I am unable to undertake too heavy/many physical tasks. I should perform light work only. For example, I easily feel tired when cooking. I have to take a break and lie down on the bed for 15 minutes. After boosting my energy, I get up and continue to cook. [from Cheng [37], 65-year-old female, breast cancer survivor] |

(*Continued*)

**Table 3.** (Continued)

| Analytical Theme | Finding | Exemplar Quotations, with participant demographic details (where available) |
|---|---|---|
| Theme four: Treatment burden as resource mismatch | Treatment and health management consumes time that could be used for other meaningful activities | "That is the biggest burden, consumption of my time. It takes me away from other activities and planning anything in my life anymore. It's become all about the hospital" [from El-Turk [64], lung cancer (age and sex missing)] |
| | | "I used to be one of those Internet junkies, trying to diagnose my own case and get my own stuff. You get tired of that after years of doing it [and] kind of just start to give in, and that's where I'm at now. . .it's just not worth my time, and the pain that it causes to try to sit at the computer to try to research stuff. . . The pain-free time is spent doing things with my kids and not getting on a computer." [from Schulman-Green [46], female with advanced breast cancer (age missing)] |
| | Cancer depletes financial resources and requires careful consideration and planning | "I had to cut back on everything, groceries, to really say, do we need that? And plan meals and make up meals out of everything." [from Timmons [75] (age, sex, and cancer type missing)] |
| | | "If I don't have any money, I can't even go out to visit people because I have to be watching that I don't run out of gasoline." [from Sleight [48], 51-year-old Latina, breast cancer survivor] |
| | | "The transport [travel to the hospital] was far. It costs three hundred thousand rupiahs [equivalent to 30 US dollars] for one trip, and six hundred thousand rupiahs [equivalent to 60 US dollars] for round-trips. . .for every treatment visit, I had to stay here for 2–3 days, and initially, I spent two hundred thousand rupiahs [approximately 20 US dollars] per night [for the accommodation]. Luckily I had a nephew here who now provided me with room to stay for free with every hospital visit." [from Prabandari [44], 58-year-old female with advanced breast cancer] |
| | Healthcare services can be difficult to access and navigate, particularly for those facing language or economic barriers | "At the time of diagnosis, I had to deal with an insensitive translator who was in a hurry and did not take the time to explain to me what cancer was. Then on my next visit, the personnel made me feel even worse, they looked at me as if I was a leper, maybe because of my poor clothing and make me feel very uncomfortable. I waited for hours, people who had appointment later than mine went in and out (. . .) all the response was that the doctor has more important things to do and that we as immigrants think that we are important but all that we are is a bunch of intruders and freeloaders." [from Ashing-Giwa [34], Latina breast cancer survivor (age missing)] |

recurrence, and the desire to maintain or re-establish personal identity/sense of normality were strong motivators for participation in the work of cancer management (see also Table 2).

## Theme three: Resource management

Individuals used personal and healthcare system resources to manage cancer. Key personal resources were time; finances; existing knowledge and skills (such as communication skills, or the ability to find, assimilate, and appraise health information); and social networks. Most studies emphasised the importance of social support for psychological well-being and for delegating certain health care tasks (exemplar quotations are in Table 3) [34, 37, 42, 44–47, 49, 54, 60, 61, 63, 65, 66, 69, 76, 79].

Another less tangible personal resource was "energy", which was a concept that was present across several studies [37, 42, 50, 63, 65, 66, 72, 79]. Energy was treated as a resource which had to be carefully managed. For example, one individual with problematic bowel symptoms and a stoma described the need to balance food intake to maintain energy, whilst trying to

control the timing of bowel movements [63]. Others managed energy levels by pacing activities or altering routines/avoiding certain tasks (see exemplar quotation, Table 3).

Healthcare system resources that were particularly valued were healthcare professional time, expertise, supportive professional relationships, and the provision of clear and tailored information. Supportive contact with healthcare professionals was reassuring.

### Theme four: Treatment burden as resource mismatch

Individuals were burdened by health care when there was a mismatch between the need for a specific resource and the availability or accessibility of that resource. The most prominent negative consequence of treatment workload (see Table 2 and exemplar quotations, Table 3) was that time expended undertaking health care work took away time available for meaningful family or personal activities/time spent with support networks [34, 46, 64, 68, 73].

Several studies [34, 44, 47, 48, 67, 75] discussed the impact of financial resources being depleted due to cancer. Paying for cancer-related expenses meant that some individuals were struggling to afford necessities, were in debt, had lost savings, or were experiencing worry. Balancing finances and managing material resources took time and energy (exemplar quotations, Table 3).

Healthcare services contributed to burden when they were difficult to access or navigate. Those from minority ethnic groups [34, 35] or experiencing language barriers, and individuals dealing with economic hardship [48, 75] faced particular barriers to accessing healthcare services (exemplar quotations are in Table 3).

Healthcare services also contributed to burden when individuals perceived that their healthcare professional lacked knowledge of a particular problem or was dismissive of their concerns [39, 49, 71, 78]. The idea of *abandonment* by the healthcare system was mentioned by participants in a number of studies [32, 39, 54, 60, 71], particularly at transition points in care such as discharge after an operation or discharge from specialist services.

### The influence of cancer type and stage

There were many similarities in treatment burden across cancer types but also some differences. Severe and persistent symptoms contributed to treatment burden. Individuals with ongoing bowel symptoms, incontinence or stomas after colorectal cancer described embarrassment and social isolation [53, 55, 60, 61, 63], resulting in limited help and support from their social networks. Stigma, embarrassment, and self-blame were prominent findings in studies of individuals with lung cancer who smoked [64, 66, 68], and there were examples of individuals becoming socially isolated due to the perception of being judged negatively by peers [68].

Individuals with stomas, incontinence, and ongoing bowel symptoms after colorectal cancer discussed the practical difficulties of integrating physical exercise into their lives, despite acknowledging this as an important aspect of survivorship care [60, 61].

The perception of "abandonment" and lack of support from the health system was mentioned frequently in studies of breast cancer survivors with lymphoedema [39, 71, 78]. There was a sense that clinicians treating breast cancer were focused on improving prognosis, lacked knowledge of lymphoedema management, and under-estimated the negative impact of lymphoedema and its management on breast cancer survivors' quality of life.

The finding that time spent on health care related activities was a source of burden was mainly noted in studies of patients with advanced and poor prognosis cancers [46, 64, 73]. Individuals who were aware of having limited life expectancies valued meaningful activities and perceived the time they spent on health care to be more burdensome.

## Quality of included studies

Most studies were rated moderately good according to the modified CASP checklist (see S3 Table). A frequent finding was that the researcher had not reflected on their role or position and how this might have influenced their research. Some studies did not give detailed descriptions of individual participant demographics, for example, few studies listed individuals' socio-economic status, and some quotations were not labelled with the demographic details of the individual who made the statement. Some papers provided a lot of raw data and participant quotations with little attempt to synthesise this data within the paper. Three papers [50, 63, 66] did not make an explicit statement about receipt of ethical approval.

# Discussion

## Main findings

Living with and beyond cancer involves cognitive, practical, and relational work for patients and necessitates the use of a range of personal and healthcare system resources. Cancer management is often perceived positively, in terms of taking control, engaging in practices that could extend life or prevent recurrence, improving psychological well-being and returning to "normality". Treatment burden occurs when resources are consumed, unavailable or difficult to access.

Chronic symptoms that persist after cancer treatment, such as bowel symptoms, lymphoe-dema, pain and fatigue contribute to treatment burden. Individuals who must carefully balance financial resources and those without strong social support networks or who have become isolated by their disease and its treatments are also particularly at risk of treatment burden. For those with advanced disease, time becomes more precious and is a valued resource. The consumption of time by health workload can be perceived as a burden.

Health system factors can contribute to perceptions of treatment burden when there are barriers to access, or when services do not meet individuals' needs. Those facing language and economic barriers seem to be particularly at risk. The notion of "abandonment" at transition points in care emphasises that individuals require expert support with the work of self-management. The shift in responsibility for health care workload from professionals to patients at transition points may lead to patient burden.

## Comparison with existing literature

Several theories and conceptual models have been used by researchers to better understand treatment burden across a range of conditions [4, 7, 8, 80, 81]. Our analysis involved a thematic approach that was not theory driven, however our findings do fit well with existing theory. Normalisation Process Theory (NPT) [81] identifies factors that facilitate or prevent the incorporation of interventions into everyday life and has been used to understand how patients integrate illness management work into their lives. NPT describes coherence work, relational work, enacting work and reflecting work. These categories align closely with the cognitive, relational, and practical work outlined in this review.

The Theory of Patient Capacity describes the factors that can influence a person's ability to manage their health [80]. This includes the biographical disruption associated with illness, the mobilisation of material and personal resources, and social functioning. Additionally, the realization of work is described as a driver for feeling successful and increasing confidence, and the environment in which health is managed is cited as an important influencer of patient capacity [80]. These factors align with our themes 'context, motivation and reward' and 'resource management'.

The Cumulative Complexity Model explains how the balance between workload and capacity can influence outcomes, with illness and treatment burden increasing if workload

outweighs capacity [7]. This aligns with our theme 'treatment burden as resource mismatch'. We found that managing resource challenges (e.g., making trade-offs in energy used or pain endured to perform an activity, or deciding how to use finite financial resources) was a major source of work for individuals living with and beyond cancer.

Burden of Treatment Theory models the relationship between the patient, their social networks and health services [4]. This theory is echoed across all four of our themes, with cancer patients working hard to mobilize resources and interact with both their social networks and healthcare providers to perform the tasks of health management.

One strong theme in the cancer literature which does not seem to be as strongly echoed in reports of treatment burden in other diseases is the sense of fear associated with cancer. Fear of death and fear of recurrence were prominent in most of the studies reviewed, and self-management work takes place in this context. Indeed, individuals' perceptions of severity and threat to life in cancer may serve as strong motivators for undertaking cancer-related health management. This may be one explanation for the mainly positive perceptions of the benefits of self-management activities (Table 2).

Multimorbidity contributes to treatment burden [82] but the management of comorbidities as a source of treatment burden did not feature as a prominent finding in any of the 48 studies reviewed here. This is surprising because over 75% of people with cancer have at least one other chronic medical condition [83] and most of the studies in this review set out with broad aims such as examining "challenges", "concerns", and "experiences" in individuals with cancer (see S2 Table). This review focused on cancer as an index condition, and many of the participants were experiencing significant ongoing cancer-related illness burden. Participants and researchers may have chosen to focus on cancer management in isolation.

Two reviews have specifically examined multimorbidity management in individuals with cancer [84, 85]. Cavers et al. [84] found that multimorbidity could increase the complexity of medicines management. Corbett et al. [85] found that the combination of old age and multimorbidity complicated self-management after cancer, and that older individuals prioritised the management of the health condition which was having the greatest negative impact on independent living [85]. It seems likely that multimorbidities compete for and consume finite personal resources and contribute to the "resource mismatch" that could lead to treatment burden in individuals with cancer.

A key finding in this review is that consumption of patients' time can lead to treatment burden. There is increasing recognition of "time toxicity" in cancer care, which is conceptualised as time spent coordinating care and in frequent visits to healthcare facilities [86].

Some healthcare systems have introduced patient navigation and nurse case manager roles in which a dedicated person or team helps patients to navigate complex systems, coordinates care, and helps patients to manage transitions in care [87]. Navigators and case managers can improve patient satisfaction and measures of patient experience with their care [87, 88]. Evidence about their effectiveness on reducing hospitalisation, use of emergency services and overall cost effectiveness is less clear [87, 89]. This review suggests that coordinating care and visiting healthcare facilities form only a small part of the work of managing cancer. Patients' time is consumed by cognitive, relational, and other practical processes related to cancer management. Many of these processes take place away from healthcare settings.

## Strengths and limitations

The inclusion of 48 original research studies with over 1000 individual participants from 11 different countries allowed for a detailed exploration of the subject and provided new insights into treatment burden in cancer. The synthesis goes beyond a simple aggregation or summary

of existing literature, developing key analytical themes and concepts which were consistently and repeatedly identified across different cancer types and in different settings. The use of at least two independent reviewers for screening and data extraction added rigour.

There were limitations to the review. Treatment burden is a relatively new concept in cancer care, and only one study was identified that specifically set out to examine treatment burden after cancer [64]. The results of this review are derived from heterogeneous studies, with wide ranging aims and scope. There is a risk that results and participant quotations from these studies have been taken out of context. However, the use of a specific definition of treatment burden at the outset of the review, along with strict inclusion and exclusion criteria, ensured that all the included studies examined treatment burden. It was also reassuring that the same themes were present across diverse studies.

Breast and colorectal cancer were over-represented compared to lung and prostate cancer in this review. There may be specific aspects of prostate and lung cancer self-management work that have not been fully explored here. Furthermore, it was beyond the scope of this review to explore caregiver perceptions of treatment burden. Caregiver experiences are likely to give additional insights into treatment burden. This is an important avenue for further enquiry.

## Implications for research and practice

Most of the resources needed to manage cancer, from finances to social networks and individual "energy", are not fixed and are likely to change over time. The demands placed on patients and their families are also likely to change during different stages in the disease and treatment trajectory. Importantly, many of the factors identified in this review that contribute to treatment burden are potentially modifiable.

Healthcare professionals should be aware of treatment burden and should consider carefully what they are asking or expecting patients to do. There may be simple ways of reducing patient/caregiver workload, for example, conducting some consultations remotely (to reduce travel time), rationalising medication dosing regimens, and reducing the administrative burden of health care for patients. Formal, professional psychological support that addresses fear associated with cancer, and support with behaviour change could also reduce the cognitive workload of cancer self-management.

Financial toxicity during cancer care is likely to vary based on the levels of social inequality in a country, and in Government provision for health insurance and welfare benefits. It may be beyond individual health care practitioners to be able to influence welfare and economic policy, but it is essential that practitioners are aware that patients who face financial pressures during cancer treatment experience treatment burden, and that treatment burden can modify important health outcomes. Professionals should specifically ask about financial burden related to disease management and be able to sign post to relevant local resources.

A careful balance needs to be struck between the supportive, reassuring nature of frequent contact between patients and the healthcare system with the burden of time spent on health care. Data from this review suggest that individuals with distressing symptoms or symptoms that limit quality of life appreciate support from health care professionals, particularly during transition points in care, such as discharge from specialist clinic follow up. There may be an important role for planned review by specialist nurses and primary care practitioners around these transition points to minimise the sense of abandonment. Conversely, individuals with poor prognosis cancer consider time as a precious and scarce resource and professionals should consider how they might limit the demands made on patients' time for health-related tasks. Future research might quantify the time spent on health care and the factors that

influence this (e.g., rurality, multimorbidity, cancer type/stage/treatment received). Quantifying "time toxicity" could be an important step in understanding and reducing treatment burden after cancer.

## Conclusions

Cancer management involves cognitive, practical and relational work for patients. Self-management work can be empowering and can give patients a sense of control over symptoms and disease outcomes. However, patients can become burdened by treatment when there is a mismatch between the personal and healthcare resources needed for health care work and their availability. Patients with limited financial resources, those who face barriers to accessing health system resources, and those with competing demands on their time may be particularly burdened by treatment. Treatment burden could be an important mediator of inequities in cancer outcomes.

## Supporting information

**S1 Table. Enhancing transparency in reporting the synthesis of qualitative research: ENTREQ checklist.**
(DOCX)

**S2 Table. Descriptions of the 48 original qualitative research articles included in the review.**
(DOCX)

**S3 Table. Modified CASP checklist used to quality assess the included articles.**
(DOCX)

**S1 Appendix. Medline search strategy.**
(DOCX)

## Acknowledgments

The authors would like to thank Melanie Bickerton, Information Consultant for Medicine and Biomedical Sciences for her assistance with database search strategies.

## Author Contributions

**Conceptualization:** Rosalind Adam.

**Data curation:** Revathi Nair, Lisa F. Duncan, Esyn Yeoh, Joanne Chan, Vaselisa Vilenskaya.

**Formal analysis:** Rosalind Adam, Revathi Nair, Lisa F. Duncan, Esyn Yeoh, Joanne Chan, Vaselisa Vilenskaya, Katie I. Gallacher.

**Funding acquisition:** Rosalind Adam.

**Investigation:** Rosalind Adam.

**Methodology:** Rosalind Adam, Katie I. Gallacher.

**Project administration:** Rosalind Adam, Lisa F. Duncan.

**Supervision:** Rosalind Adam.

**Writing – original draft:** Rosalind Adam.

**Writing – review & editing:** Rosalind Adam, Revathi Nair, Lisa F. Duncan, Esyn Yeoh, Joanne Chan, Vaselisa Vilenskaya, Katie I. Gallacher.

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
