## [Decision Letter · Decision Letter 0]

8 Mar 2023

PONE-D-22-21252Treatment burden in individuals living with and beyond cancer: a systematic review of qualitative literaturePLOS ONE

Dear Dr. Adam,

Thank you for submitting your manuscript to PLOS ONE. After careful consideration, we feel that it has merit but does not fully meet PLOS ONE’s publication criteria as it currently stands. Therefore, we invite you to submit a revised version of the manuscript that addresses the points raised during the review process.

We look forward to receiving your revised manuscript.

Kind regards,

Federica Canzan

Academic Editor

PLOS ONE

Journal Requirements:

2. Please amend the manuscript submission data (via Edit Submission) to include authors Revathi Nair, Esyn Yeoh, Joanne Chan, Vaselisa Vilenskaya & Katie I Gallacher.

Reviewers' comments:

Reviewer's Responses to Questions

**Comments to the Author**

1. Is the manuscript technically sound, and do the data support the conclusions?

Reviewer #1: Partly

Reviewer #2: Yes

2. Has the statistical analysis been performed appropriately and rigorously? 

Reviewer #1: Yes

Reviewer #2: Yes

3. Have the authors made all data underlying the findings in their manuscript fully available?

Reviewer #1: Yes

Reviewer #2: Yes

4. Is the manuscript presented in an intelligible fashion and written in standard English?

Reviewer #1: No

Reviewer #2: Yes

5. Review Comments to the Author

Reviewer #1: The content and methods described in the analysis are sound. What concerns me is the layout of the manuscript and the typographical errors found. For example Table S1 as opposed to Table 1. The theming names chosen for this qualitative systematic review also lack creativity and could be further refined to be more engaging and attract the readership. Also, the in-text citation numbers could be refined for example (65,66,67) can be identified in the manuscript as (65-67). Also, according to author guidelines, the citations are to be encapsulate by [ ] and not parentheses. Since, PLOS ONE does not copyedit accepted manuscripts, I cannot recommend the publication as is be accepted. There are other significant errors noted throughout the manuscript.

Reviewer #2: Thank you for the opportunity to review this manuscript. The authors present a systematic review of qualitative literature on treatment burden in individuals living with and beyond cancer.

This type of research has been undervalued in the past and I enjoyed reading the materials.

I have a couple of questions which may help to further improve the manuscript:

Abstract:

-'This work' is a bit odd in the context. Perhaps refer to health management instead?

-Add energy in the final sentence

Introduction:

-I recommend outlining that you are not focused on palliative care/end of life phase.

Methods:

-Ideally, full texts were judged by at least 2 reviewers: please explain

-Why did you select these cancer types?

Results:

-Theme one: please add stoma stigma as well

-What about hair loss?

-What about sexual function/menopause/relationships?

Discussion:

-What can HCP do to avoid abandonment? Create care paths, specialised nurses?

-What can HCP do to avoid worry? And what can institutes/governments do about financial worry/support?

-Are the health care systems not creating bias in your literature selection? If privately insured/rich, one can pay for all support?

-Support systems often stop after surgery: the tumor is gone, right? What can we do to support patients and prepare them for this 'loss'?

Tables are clear and concise.

6. PLOS authors have the option to publish the peer review history of their article (what does this mean?). If published, this will include your full peer review and any attached files.

Reviewer #1: No

Reviewer #2: No

---

## [Author Response · Author response to Decision Letter 0]

22 Mar 2023

POINT BY POINT RESPONSES

REVIEWER ONE

Comment: The content and methods described in the analysis are sound.

Response: Thank you. 

Comment: What concerns me is the layout of the manuscript and the typographical errors found. For example Table S1 as opposed to Table 1.

Response: We have carefully proof-read the article again. Please note that Table S1 refers to Supplementary or Supporting Information Table 1, rather than Table 1, which is contained within the manuscript itself. We note that the convention for PLOS One is to refer to supporting information files as “S1 Table” rather than “Table S1” so supporting information files have been renamed to reflect this. We have also added a list of Supporting information captions at the end of the manuscript file. 

Comment: The theming names chosen for this qualitative systematic review also lack creativity and could be further refined to be more engaging and attract the readership.

Response: Thank you for this suggestion. The themes were less of a product of creative thinking and more of an inductive analysis, whereby the data determined the themes. Six authors reviewed the study data, and the themes were refined and tested against the original data during the analysis so that they remained true to the findings of the original research, but also added new insights into the phenomenon of treatment burden after cancer (see Figure 1). We did not use any one theory to guide our data analysis and were reassured when we found that our themes fitted well with existing theories of treatment burden and human behaviour (see discussion, comparison with existing literature). Since submitting this article, we have also presented our findings to different audiences, including clinicians and health psychologists, and are reassured that the themes seem to resonate with professionals from different backgrounds. 

Comment: The in-text citation numbers could be refined for example (65,66,67) can be identified in the manuscript as (65-67).

Response: Thank you. We used reference management software for citations and have manually checked that citation numbers and formatting to ensure that these are concise and correct.

Comment: The citations are to be encapsulate by [ ] and not parentheses.

Response: These have been changed.

Comment: There are other significant errors noted throughout the manuscript.

Response: We carefully proof-read the article and did not find significant errors. To resolve any discrepancies, we enlisted a senior Professor, not involved in this review to carefully proof-read our manuscript. He noted two words that we might change in the abstract for clarity, but no other typographical or grammatical errors. We would be happy to fix any errors if these can be pointed out individually. 

REVIEWER TWO

Comment: This type of research has been undervalued in the past and I enjoyed reading the materials.

Response: Thank you.

Comment: Abstract: “this work” is a bit odd in the context. Perhaps refer to health management instead?

Response: We have changed the line to read “health management work”, which improves the flow. We have referred to “work” throughout the paper because this fits with conventional definitions of treatment burden and with literature dating back to the 1980s when Corbyn and Strauss compared illness management to “work”. In our results, we found that health management was compared with or literally described as “work” (Theme one). 

Comment: Abstract: add energy in the final sentence

Response: We have amended the final sentence to add “energy”. 

Comment: Introduction: I recommend outlining that you are not focused on palliative/end of life phase

Response: We included studies of patients with a confirmed diagnosis of cancer at all stages in the disease trajectory, including those who had completed potentially curative treatment, and individuals with advanced cancers. In our inclusion criteria (see Table 1), we include “Individuals with a confirmed diagnosis of colorectal, lung, prostate, or breast cancer, at any point in disease and treatment trajectory.” We did not exclude studies focusing on palliative care. In the review, there are seven papers that focus specifically on advanced cancers (see also S2 Table):

• Hall et al. include individuals with stage III and IV lung cancer and melanoma

• Prabandari et al include individuals with metastatic breast cancer

• Schulman-Green et al. focus on individuals with metastatic breast cancer who have failed first-line therapy

• Shih et al focus on the views of individuals with stage IV adenocarcinoma of the lung

• Walshe et al include individuals with “advanced” stage 3-4 breast, prostate, lung, or colorectal cancer who are in receipt of palliative care

• Webber et al. focus on breakthrough pain in advanced cancer

• Zhang et al. include individuals with advanced (stage 3 or 4) lung cancer.

We are confident that these papers encompass views and opinions of individuals undergoing palliative care. One of the interesting findings in our review was that the burden of time spent on treatment seems to be particularly prominent in those with poor-prognosis cancers. These findings are encapsulated in quotations from individuals with lung cancer and metastatic breast cancer (Table 3, theme 4), and are specifically brought out in the results section (heading “The influence of cancer type and stage”):

“The finding that time spent on health care related activities was a source of burden was mainly noted in studies of patients with advanced and poor prognosis cancers (48,65,74). Individuals who were aware of having limited life expectancies valued meaningful activities and perceived the time they spent on health care to be more burdensome.”

The “end of life phase” has a number of definitions and meanings. If the reviewer is referring to changes in the last hours and days of life such as drowsiness and difficulties communicating, these are likely to preclude participation in a qualitative interview study. Nevertheless, we did not actively exclude papers that might have included individuals coming towards the end of their life.

Comment: Methods: Ideally, full texts were judged by at least two reviewers: please explain

Response: We included multiple authors at all stages of the review process, from study identification, applying inclusion/exclusion criteria, extracting data, analysing the data, and quality assessing the papers. Independent study assessment by multiple authors added methodological rigor and reduced the chance that we would miss important papers/data, or that the views of one author would introduce bias into our results. We have made this clear in the paper as follows. In the methods section, under “identification of studies” we state that: 

“Titles and abstracts were reviewed independently by two authors (RA, JC, AK, EY, and VV). Full texts were retrieved for all studies which were judged by at least one author to be potentially eligible for inclusion. Full texts were reviewed independently by at least two authors and any disagreements were resolved by discussion with a third author.”

We also state under “data extraction and synthesis” that: 

“Data extraction was performed independently by at least two authors, and three authors (RA, LD, and AK) met to agree the final data extraction form for each study. A unified document containing the data extracted from all studies was circulated to the whole review team.”

Under “quality assessment”, we state that:

“Study quality assessment was undertaken independently by two authors (JC and EY) using a modified CASP checklist (32). Areas of uncertainty or disagreement were resolved by discussion with a third author (LD).”

Comment: Methods: why did you select these cancer types?

Response: We state in the last sentence of the introduction that “The review focuses on breast, prostate, colorectal, and lung cancers, which are the most common cancers globally (25), and encompass a range of symptoms, prognoses, treatment modalities, and late effects.” We have changed the wording of this sentence to state that we focus on these cancers “because they” are the most common cancers globally and encompass a range of symptoms, prognoses, treatment modalities, and late effects. In our review, we identified 14,730 titles from database searching for these four cancer types. To have included all cancer types in our search would have created too many database records to have been managed systematically, even by a review team of seven authors. We are confident that including the four most common cancers will encompass a broad and representative range of patient experiences.

Comment: Results: Theme one: please add stoma stigma as well

Response: We have amended the text in theme one to include the following:

“Many tasks involved a combination of practical and cognitive work, for example, concealing a stoma might involve planning what to wear, and then the practical tasks of using tape to mask the stoma. Several studies described participants making efforts to avoid the topic of cancer being raised by others during social or daily activities. Individuals were motivated to conceal physical signs of their cancer treatment to maintain a sense of identity and normality, to maintain their sense of physical attractiveness, and to mitigate social stigma or judgements about having cancer.”

Comment: Results: what about hair loss? What about sexual function/menopause/relationships?

Response: We did not present a comprehensive overview of the symptoms and illness-related problems experienced by interview participants. Previous reviews have focused in detail on illness burden in cancer, including details of distressing symptoms such as erectile dysfunction after prostate cancer treatment, menopausal symptoms with hormonal treatments or after chemotherapy, and hair loss. A subtle difference in our review is that we focused on the work required to manage disease and symptoms, rather than individual problems or symptoms. In theme two we note that work is required to maintain relationships. We have added extra detail in this theme as follows:

“Relational work for those with cancer involved enlisting the support of others (such as healthcare professionals, friends, and significant others) to assist with the work of health management. Many studies [35,38,42,48,50,61,64,66,70,77,79] noted that cancer and cancer treatment altered the dynamics of personal relationships by reducing sexual intimacy or changing roles within relationships. For example, friends and loved ones had to take on caregiving roles, and individuals with cancer struggled to fulfil the family responsibilities they had valued prior to their diagnosis (e.g., looking after children, helping with household chores). It took significant effort to maintain important relationships during cancer treatment and beyond.”

Comment: Discussion: What can HCP to avoid abandonment? Create care paths, specialised nurses? What can HCP do to avoid worry? And what can institutes/governments do about financial worry/support? 

We have made the following additions to the “implications for research and practice” sections of our paper:

“Healthcare professionals should be aware of treatment burden and should consider carefully what they are asking or expecting patients to do. There may be simple ways of reducing patient/caregiver workload, for example, conducting some consultations remotely (to reduce travel time), rationalising medication dosing regimens, and reducing the administrative burden of health care for patients. Formal, professional psychological support that addresses fear associated with cancer, and support with behaviour change could also reduce the cognitive workload of cancer self-management. 

Financial toxicity during cancer care is likely to vary based on the levels of social inequality in a country, and in Government provision for health insurance and welfare benefits. It may be beyond individual health care practitioners to be able to influence welfare and economic policy, but it is essential that practitioners are aware that patients who face financial pressures during cancer treatment experience treatment burden, and that treatment burden can modify important health outcomes. Professionals should specifically ask about financial burden related to disease management and be able to sign post to relevant local resources.

A careful balance needs to be struck between the supportive, reassuring nature of frequent contact between patients and the healthcare system with the burden of time spent on health care. Data from this review suggest that individuals with distressing symptoms or symptoms that limit quality of life appreciate support from health care professionals, particularly during transition points in care, such as discharge from specialist clinic follow up. There may be an important role for planned review by specialist nurses and primary care practitioners around these transition points to minimise the sense of abandonment. Conversely, individuals with poor prognosis cancer consider time as a precious and scarce resource and professionals should consider how they might limit the demands made on patients’ time for health-related tasks. Future research might quantify the time spent on health care and the factors that influence this (e.g. rurality, multimorbidity, cancer type/stage/treatment received). Quantifying “time toxicity” could be an important step in understanding and reducing treatment burden after cancer.”

Comment: Discussion: Are the health care systems not creating bias in your literature selection? If privately insured/rich, one can pay for all support?

Response: We included 48 studies from 13 different countries, with over 1000 individual participants, encompassing a variety of different models of health care funding and provision. We identified and selected studies systematically and did not make any inclusions/exclusions based on health system funding/organisation, or participant demographics. We are confident that the studies included have been able to represent individuals facing financial difficulties, as well as those who were able to pay for support. For example, Ashing-Giwa et al investigated the experiences of 102 women from different ethnicities (African American, Asian American, Latina and Caucasian) and examined the challenges faced by these women in the context of the healthcare system in the United States of America (see S1 table). A study by Timmons et al found that individuals in Ireland faced significant financial burden after cancer. Ireland has a funded public health system. Financial burden is likely to be an important consideration for individuals living with and beyond cancer worldwide.

Comment: Tables are clear and concise.

Response: Thank you.

---

## [Decision Letter · Decision Letter 1]

11 Apr 2023

PONE-D-22-21252R1Treatment burden in individuals living with and beyond cancer: a systematic review of qualitative literaturePLOS ONE

Dear Dr. Adam,

Thank you for submitting your manuscript to PLOS ONE. After careful consideration, we feel that it has merit but does not fully meet PLOS ONE’s publication criteria as it currently stands. Therefore, we invite you to submit a revised version of the manuscript that addresses the points raised during the review process. One reviewer risen a minor comment that can be addressed. 

We look forward to receiving your revised manuscript.

Kind regards,

Federica Canzan

Academic Editor

PLOS ONE

Journal Requirements:

Reviewers' comments:

Reviewer's Responses to Questions

**Comments to the Author**

1. If the authors have adequately addressed your comments raised in a previous round of review and you feel that this manuscript is now acceptable for publication, you may indicate that here to bypass the “Comments to the Author” section, enter your conflict of interest statement in the “Confidential to Editor” section, and submit your "Accept" recommendation.

Reviewer #3: (No Response)

2. Is the manuscript technically sound, and do the data support the conclusions?

Reviewer #3: Yes

3. Has the statistical analysis been performed appropriately and rigorously? 

Reviewer #3: N/A

4. Have the authors made all data underlying the findings in their manuscript fully available?

Reviewer #3: Yes

5. Is the manuscript presented in an intelligible fashion and written in standard English?

Reviewer #3: Yes

6. Review Comments to the Author

Reviewer #3: The article is well written and it addresses an important topic. I do not see any outstanding issues as regarding the methodology of the study and the article structure. I have just a few minor comments.

However, as one the previous reviewers pointed out, I do not completely agree with the term "work" used by the authors throughout the manuscript. Specifically in some cases as for example at page 17 I would substitute "Work was time-consuming and included cognitive work, practical work, and relational work." with "the cancer experience was time-consuming and included cognitive work, practical work, and relational work." This appears also at page 23 in "The influence of cancer type and stage chapter", where "Severe and persistent symptoms could make health care work more difficult". The literature talks extensively of the difficult self-management and burden of cancer patients and in line with the literature I would change the term "work" with "management".

Discussion - main findings:

The notion of abandonment and shift in responsability is not discussed taking into account the figure of the nurse case manager, which exists in certain healthcare systems and can help to smooth the transition at the end of cancer treatment, in the new phase of the disease trajectory survivorship.

7. PLOS authors have the option to publish the peer review history of their article (what does this mean?). If published, this will include your full peer review and any attached files.

Reviewer #3: No

---

## [Author Response · Author response to Decision Letter 1]

20 Apr 2023

We have included a full point-by-point response to reviewer comments.

---

## [Decision Letter · Decision Letter 2]

15 May 2023

Treatment burden in individuals living with and beyond cancer: a systematic review of qualitative literature

PONE-D-22-21252R2

Dear Dr. Adam,

We’re pleased to inform you that your manuscript has been judged scientifically suitable for publication and will be formally accepted for publication once it meets all outstanding technical requirements.

Kind regards,

Federica Canzan

Academic Editor

PLOS ONE

Additional Editor Comments (optional):

Reviewers' comments:

Reviewer's Responses to Questions

**Comments to the Author**

1. If the authors have adequately addressed your comments raised in a previous round of review and you feel that this manuscript is now acceptable for publication, you may indicate that here to bypass the “Comments to the Author” section, enter your conflict of interest statement in the “Confidential to Editor” section, and submit your "Accept" recommendation.

Reviewer #3: All comments have been addressed

2. Is the manuscript technically sound, and do the data support the conclusions?

Reviewer #3: Yes

3. Has the statistical analysis been performed appropriately and rigorously? 

Reviewer #3: N/A

4. Have the authors made all data underlying the findings in their manuscript fully available?

Reviewer #3: Yes

5. Is the manuscript presented in an intelligible fashion and written in standard English?

Reviewer #3: Yes

6. Review Comments to the Author

Reviewer #3: (No Response)

7. PLOS authors have the option to publish the peer review history of their article (what does this mean?). If published, this will include your full peer review and any attached files.

Reviewer #3: No

---

## [Editor Report · Acceptance letter]

17 May 2023

PONE-D-22-21252R2 

Treatment burden in individuals living with and beyond cancer: a systematic review of qualitative literature 

Dear Dr. Adam:

I'm pleased to inform you that your manuscript has been deemed suitable for publication in PLOS ONE. Congratulations! Your manuscript is now with our production department. 

Kind regards, 

on behalf of

Professor Federica Canzan 

Academic Editor

PLOS ONE